# Impact of vaccination on infection or death from COVID-19 in individuals with laboratory-confirmed cases: Case-control study

Porfirio Felipe Hernández Bautista[1☯], Concepción Grajales Muñiz[1‡], David Alejandro Cabrera Gaytán[1☯*], Teresita Rojas Mendoza[1‡], Alfonso Vallejos Parás[2☯‡], Clara Esperanza Santacruz Tinoco[1‡], Julio Elias Alvarado Yaah[1‡], Yu Mei Anguiano Hernández[1‡], Nancy Sandoval Gutiérrez[1‡], Leticia Jaimes Betancourt[3‡]

1 Instituto Mexicano del Seguro Social, Coordinación de Calidad de Insumos y Laboratorios Especializados, Coahuila, Mexico, 2 Instituto Mexicano del Seguro Social, Coordinación de Vigilancia Epidemiológica, Coahuila, Mexico, 3 Instituto Mexicano del Seguro Social, Unidad de Medicina Familiar No. 7., Coahuila, Mexico

☯ These authors contributed equally to this work.
‡ CGM, TRM, AVP, CEST, JEAY, YMAH, NSG and LJB also contributed equally to this work.
* david.cabrerag@imss.gob.mx, dcpreventiva@gmail.com

**Data Availability Statement:** All relevant data are within the paper and its Supporting information files.

## Abstract

### Objectives

The objective of this study is to estimate the effectiveness of COVID-19 vaccines in people treated within the social security system whose vaccination status was reported to the epidemiological surveillance system.

### Study design

Case-control study.

### Methods

This was a case-control study conducted. The records of individuals with suspected cases of COVID-19 registered in the epidemiological surveillance system between February 1 and June 30, 2021, were studied. RT–qPCR was performed to determine SARS-CoV-2 infection; those with a positive result were considered cases, and those with a negative result were considered controls. The ratio between cases and controls was 1:1.3. The crude and adjusted vaccine effectiveness was considered the prevention of symptomatic infection and death and calculated as the difference between the dose and the risk, with a survival analysis among vaccinated people.

### Results

A total of 94,416 individuals were included, of whom 40,192 were considered cases and 54,224 controls; 3,781 (4.00%) had been vaccinated against COVID-19. Vaccination also proved to be a protective factor against COVID-19, especially in the population who received a second dose (OR = 0.31; 95% CI 0.28–0.35). With the application of the vaccine,

**Funding:** The authors received no specific funding for this work.

**Competing interests:** The authors have declared that no competing interests exist.

there was a protective effect against mortality (OR = 0.76; 95% CI 0.66–0.87). Disease prevention was higher for the BNT162-2 mRNA vaccine (82%) followed by the ChAdOx1 vaccine (33%). In the survival analysis, vaccination provided a protective effect.

## Conclusions

There was a positive impact of vaccines for the prevention of symptomatic COVID-19, with a second dose generating greater efficacy and a reduction in deaths.

## Introduction

An epidemic of atypical pneumonia was identified in China, later being declared a public health emergency of international importance by the World Health Organization (WHO) [1]. The agent responsible for this condition was identified as SARS-CoV-2. In Mexico, the first case was confirmed on February 27, and on April 1, an emergency was declared in the country. In Mexico, there have been three epidemic waves of COVID-19 [2]; these waves have presented changes in epidemiological behavior attributed to vaccination and government changes in restrictive measures [3–5]. Vaccination against COVID-19 began at the end of December 2020 with a federal strategy [6].

In initial studies, the efficacy of the vaccine against COVID-19 has yielded promising results [7–12] However, as more people have been vaccinated, it has been revealed that immunity may be affected not only by factors inherent to the host but also by the vaccination policies implemented and also by the prevalent epidemiological panorama in a geographic region. The Mexican strategy of the vaccination program aims to reduce the fatality rate with the maximum benefit [6], for which reason it has begun with older adults and health personnel, from there gradually and descending to other age groups. However, the combination of a high rate of virus replication within an individual and a suboptimal level of neutralizing antibodies has been documented to be the exact environment in which resistant viruses or new variants are considered likely to emerge and spread [13, 14]; therefore, the importance of an effective mass vaccination program. In this sense, BNT162b2 vaccine has stated that there is no evidence that vaccine protection from a single dose extends beyond 21 days [15]. The ChAdOx1 clinical trials did include different dose spacings, and limited evidence suggests that longer intervals (two to three months) did not affect and may even have improved vaccine efficacy [15, 16]. The epidemiological panorama of Mexico that presented in July and August of 2021 was a multitude of cases, representing the third epidemic wave, which was most evident in states located in the North of the country and on both Pacific Peninsulas, including Mexico City and the Metropolitan Area of the State of Mexico, with a cumulative incidence rate of 2,061.8 per 100,000 inhabitants [2, 17], mainly affecting those between the ages of 18 to 49 years in Mexico City [13]. Until July 16, 2021 were notified 188,655,968 cases and 4,067,517 deaths; most of the accumulated confirmed cases were found in the center of the country. The lethality rate was 2.2% (deaths with positive result). The distribution by sex in confirmed deaths shows a predominance of 62% in men and the median age at death was 64 years [2]. Anti-SARS-CoV-2 vaccination has become another control measure, such as social distancing and respiratory protection. In a report published on July 4, the Ministry of Health reported that 47,197,494 total doses had been administered and that 32,575,554 people were fully vaccinated [18]. However, social isolation measures during the third wave were different from those in the first two epidemic waves, despite being a highly transmissive epidemiological situation. The strategy of the vaccination program offers a unique opportunity to evaluate the simultaneous effectiveness and

impact of a diverse set of available COVID-19 vaccines against SARS-CoV-2 infection and COVID-19 related death, and also report estimates for effectiveness of the vaccines. Several studies reported the effectiveness and safety of the COVID-19 vaccine in the real world recently, but the results remain controversial. Therefore, the objective of this study was to evaluate the effectiveness and impact of vaccination against symptomatic infection and death in individuals with laboratory-confirmed cases listed in the epidemiological surveillance system of the Mexican Social Security Institute (IMSS).

## Material and methods

### Ethics statement

Given that it is a retrospective study without direct intervention with the selected subject or with the operation of the medical unit. All patient data were handled confidentially using only the SINOLAVE file number as part of our institutional epidemiological surveillance, ethics review and informed consent were not required by the institutional review board.

The resources were only available to the researchers for the purposes described for this study and epidemiological surveillance in the IMSS. The study analyzed a vaccine strategy implemented by IMSS, and it was not designed as a clinical trial.

This study was submitted as a research protocol to the Local Committee for Scientific Research of the IMSS and was approved with the registration number R-2022-3605-058.

### General description of the study

This was a national-based, multicenter, retrolective case-control study of confirmed cases of COVID-19 reported in the national epidemiological surveillance system called SINOLAVE, which operates in three levels of medical care of the IMSS of Mexico. Individuals who were registered from February 1 to June 30, 2021, with a history of vaccination against COVID-19 and with the type of care provided in IMSS medical units, were selected. The patients were divided in: a) individuals with laboratory-confirmed cases and b) individuals with negative RT–qPCR results. This study was based on the results published by the COVID-19 Vaccination Effectiveness Working Group in Spain [19]. In Mexico, vaccination began at the end of December 2020. Therefore, the records for that month and January 2021 were excluded.

### Operational definitions

Cases were considered individuals with a positive SARS-CoV RT–qPCR test who received care in medical units of the IMSS and who had been vaccination against COVID-19 at least 15 days before the onset of symptoms from the second dose or single dose. Controls were considered individuals with a negative SARS-CoV RT–qPCR test who received care in medical units of the IMSS and who had been vaccinated against COVID-19 at least 15 days before the onset of symptoms from the second dose.

### Data collection and management

The data were obtained from SINOLAVE, a database that contains detailed information (including sociodemographic variables, type of care, clinical manifestations, vaccination history, laboratory results and clinical outcomes) on all individuals with confirmed cases of COVID-19. The laboratory tests were conducted among the network of epidemiological surveillance laboratories in Mexico, which includes IMSS support laboratories with the capacity to perform RT–qPCR and approval by the national Secretariat of Health [20]. Data collection were collected one month later after of the court date (July 22, 2021). Vaccination against

COVID-19 was self-reported, and this information was obtained for each individual from the institutional epidemiological surveillance system.

## Classification of vaccination against COVID-19

Vaccination against COVID-19 was classified as follows: a) one dose–received the first dose with no record of receiving a second dose: b) ideal complete vaccination–received the complete vaccination schedule and the minimum period before receiving the second dose has passed based on the sectorial guidelines of the country [7–12] and the biological agent (BNT162b2 mRNA, 21 days after the first dose; ChAdOx1, 8 to 12 weeks (56 to 84 days) after the first dose; Gam-COVID-Vac, 21 days after the first dose; CoronaVac3CL, 28 days after the first dose; and Ad26 and Ad5-nCoV, single doses) and c) vaccination with two nonideal doses–two doses of a vaccine outside the time recommended between the first and second doses.

## Statistical analysis

For the univariate analysis, simple frequencies and measures of central tendency and dispersion were determined. Odds ratios (OR) were estimated by age and sex, as well as by the number of doses and type of biological for the bivariate analysis. The characteristics of the cases and controls were compared using the $X^2$ test. The association between vaccination status and baseline characteristics was estimated for both the case group and the control group. The analysis was performed by group based on the type of vaccine, age group, presence of health personnel and chronic comorbidity. In addition, a binary logistic regression model (dependent variable: positive case/death due to COVID-19) was adjusted to the variables with bivariate significance. For deaths, the data were analyzed using survival analysis methods; survival was estimated with standardized cumulative risk curves for the two groups: vaccinated and unvaccinated. The Kaplan–Meier method and Cox multivariate analysis were used to calculate cumulative risk. To determine vaccine effectiveness, the risk minus the dose was estimated. For vaccination as a preventive factor, vaccine efficacy (VE) was estimated by subtraction ((1—OR) X 100), for each biological agent and number of doses, with the 95% confidence interval [19]. The statistical analysis was performed in Epi Info version 7.

## Results

### General data

From February to July 2021, a total of 94,416 records were analyzed, of which 40,192 indicated positive laboratory-confirmed cases of COVID-19 and 54,224 indicated negative test results for COVID-19. The average age among the cases was 56 years (range 20), and that among the controls was 57 years (range = 23). Most cases were between 45 to 64 years of age (52.37%) and controls (44.27%). Smoking and occupation as health personnel were different between groups, with a predominance in controls (6.77% and 5.35%, respectively, for cases and 8.59% and 12.52%, respectively, for controls. The identified comorbidities were mainly arterial hypertension (35.59%, 33,604); diabetes mellitus (27.85%, 26,296); obesity (14.87%, 14,037); and chronic obstructive pulmonary disease (COPD) (4.07%, 3,842). The results are summarized in Tables 1 to 3. The ratio between cases and controls was 1:1.3.

COVID-19 cases had a median number of fewer days after being vaccinated compared to controls 20 vs 42, p<0.000, while in deceased and survivors of COVID-19 there was no statistically significant difference 19 vs 20, p = 0.4181.

**Table 1. Characteristics of COVID-19 cases and controls.**

| CHARACTERISTICS OF COVID-19 CASES AND CONTROLS | | | | |
|---|---|---|---|---|
| Variables | Cases | % | Controls | % |
| Man | 21,723 | 54.05 | 25,228 | 46.53 |
| Women | 18,469 | 45.95 | 28,996 | 53.47 |
| | 40,192 | | 54,224 | |
| Average age* | 56 | 20 | 57 | 23 |
| Age group | | | | |
| 18–24 years | 419 | 1.04 | 1,076 | 1.98 |
| 25–44 years | 6,288 | 15.64 | 10,669 | 19.68 |
| 45–64 years | 21,048 | 52.37 | 24,003 | 44.27 |
| 65 and more | 12,437 | 30.94 | 18,476 | 34.07 |
| Smoking | 2,723 | 6.77 | 4,656 | 8.59 |
| Health personnel | 2,149 | 5.35 | 6,787 | 12.52 |
| *Median and interquartile range | | | | |
| **Deaths and survivors, main characteristics** | | | | |
| Variables | Deaths | % | Survivors | % |
| Man | 7,340 | 59.54 | 5,967 | 54.76 |
| Women | 4,987 | 40.46 | 4,930 | 45.24 |
| Average age* | 65 | 18 | 57 | 18 |
| Age group | | | | |
| 18–24 years | 13 | 0.11 | 56 | 0.51 |
| 25–44 years | 610 | 4.95 | 1,307 | 11.99 |
| 45–64 years | 5,253 | 42.61 | 6,176 | 56.68 |
| 65 and more | 6,451 | 52.33 | 3,358 | 30.82 |
| Smoking | 921 | 7.47 | 778 | 7.23 |
| Health personnel | 91 | 0.74 | 216 | 1.98 |
| Endotracheal intubation | 1,476 | 12.06 | 124 | 1.15 |
| Admission to Intensive Care Unit | 56 | 0.45 | 23 | 0.21 |
| Onset after vaccination* | 20 | 51 | 42 | 73 |
| *Median and interquartile range | | | | |
| **Chronic illness** | | | | |
| Diseases | Deaths | % | Survivors | % |
| Arterial hypertension | 5,664 | 45.95 | 4,060 | 37.26 |
| Diabetes mellitus II | 4,309 | 30.38 | 3,311 | 34.96 |
| Obesity | 2,186 | 17.73 | 1,842 | 16.9 |
| Pulmonary tuberculosis | 10 | 0.08 | 6 | 0.06 |
| COPD | 545 | 4.43 | 317 | 2.91 |
| Asthma | 198 | 1.61 | 191 | 1.75 |
| Cardiovascular disease | 597 | 4.84 | 422 | 3.87 |
| Cancer | 106 | 0.86 | 74 | 0.68 |
| Kidney disease | 823 | 6.68 | 468 | 4.29 |
| Hepatic disease | 63 | 0.51 | 38 | 0.35 |
| Hemolytic anemia | 20 | 0.16 | 6 | 0.06 |
| Neurological disease | 59 | 0.48 | 29 | 0.27 |
| **By vaccine** | | | | |
| Vaccine | Deaths | % | Survivors | % |
| **Unvaccinated** | 11,921 | 96.71 | 10,431 | 95.72 |
| BNT162b2 mRNA | 131 | 1.06 | 131 | 1.20 |

*(Continued)*

**Table 1.** (Continued)

| | | | | |
|---|---|---|---|---|
| ChAdOx1 | 120 | 0.97 | 157 | 1.44 |
| CoronaVac3CL | 56 | 0.45 | 60 | 0.55 |
| Ad5-nCoV | 18 | 0.15 | 40 | 0.37 |
| Gam-COVID-Vac | 26 | 0.21 | 22 | 0.20 |
| BBIBP-CorV | 1 | 0.01 | 3 | 0.03 |
| Ad26 | 2 | 0.02 | 7 | 0.06 |
| Unknown | 52 | 0.42 | 46 | 0.42 |

**Table 2. Comorbility in cases and controls of COVID-19.**

| COMORBILITY IN CASES AND CONTROLS OF COVID-19 | | | | |
|---|---|---|---|---|
| Disease | Cases | % | Controls | % |
| Arterial hypertension | 14,058 | 34.98 | 19,546 | 36.05 |
| Diabetes mellitus II | 10,775 | 26.81 | 15,521 | 28.62 |
| Obesity | 6,619 | 16.47 | 7,418 | 13.68 |
| Pulmonary tuberculosis | 29 | 0.07 | 87 | 0.16 |
| COPDS | 1,131 | 2.82 | 2,711 | 5.00 |
| Asthma | 681 | 1.69 | 1,385 | 2.55 |
| Cardiovascular disease | 1,330 | 3.31 | 3,393 | 6.26 |
| Cancer | 256 | 0.64 | 751 | 1.38 |
| Kidney disease | 1,632 | 4.06 | 4,520 | 8.34 |
| Hepatic disease | 124 | 0.31 | 372 | 0.69 |
| HIV infection | 129 | 0.32 | 321 | 0.59 |
| Hemolytic anemia | 38 | 0.09 | 113 | 0.21 |
| Neurological disease | 110 | 0.27 | 352 | 0.65 |

**Table 3. Type of vaccine in cases and controls of COVID-19.**

| Vaccine | Cases | % | Controls | % |
|---|---|---|---|---|
| **Unvaccine** | 38,844 | 96.65 | 51,345 | 94.69 |
| BNT162b2 mRNA | 442 | 1.10 | 1,634 | 3.01 |
| ChAdOx1 | 403 | 1.00 | 478 | 0.88 |
| CoronaVac3CL | 190 | 0.47 | 270 | 0.50 |
| Ad5-nCoV | 85 | 0.21 | 89 | 0.16 |
| Gam-COVID-Vac | 75 | 0.19 | 83 | 0.15 |
| BBIBP-CorV | 8 | 0.02 | 10 | 0.02 |
| Ad26 | 11 | 0.03 | 3 | 0.01 |
| Unknown | 134 | 0.33 | 312 | 0.58 |

## Vaccine efficacy

A total of 3,781 patients (4.00%) had been vaccinated. The following vaccines were administered: BNT162b2 mRNA (2,076 doses), ChAdOx1 (881 doses), CoronaVac3CL (460 doses), Ad5-nCoV (174 doses), Gam-COVID-Vac (158 doses), BBIBP-CorV (18 doses) and Ad26 (14 doses). In total, 46.7% (1,768) had received the full schedule at the time of data collection.

**Table 4. Factors to get sick from COVID-19.**

| Variables | Cases | Controls | OR | CI95% |
|---|---|---|---|---|
| Man | 21,723 | 25,228 | 1.35 | 1.31–1.39 |
| Women | 18,469 | 28,996 | 1 | - |
| Age group | | | | |
| 18–24 years | 419 | 1,076 | 1 | - |
| 25–44 years | 6,288 | 10,669 | 1.51 | 1.34–1.70 |
| 45–64 years | 21,048 | 24,003 | 2.25 | 2.01–2.52 |
| 65 and more | 12,437 | 18,476 | 1.72 | 1.54–1.94 |
| Smoking | 2,723 | 4,656 | 0.77 | 0.73–0.81 |
| Health personnel | 2,149 | 6,787 | 0.39 | 0.37–0.41 |

Based on the crude OR, the following was a risk factor for symptomatic COVID-19: 45 to 64 years of age (OR = 2.25; 95% CI 2.01–2.52). A preventive factor was being a health worker (OR = 0.39; 95% CI 0.37–0.41), as were most comorbidities, for example, renal pathology OR = 0.46 (95% CI 0.43–0.49) or liver pathology OR = 0.44 (95% CI 0.36–0.54), except for systemic arterial hypertension, diabetes mellitus and obesity (Table 4).

Vaccination was a protective factor against COVID-19, especially among the population with a second dose (OR = 0.31, 95% CI 0.28–0.35). BNT162-2 mRNA provided the greatest protection, even from the first dose (OR = 0.61, 95% CI 0.53–0.70), with the risk decreasing further after the second dose (OR = 0.18; 95% CI 0.15–0.21). For ChAdOx1, a protective effect was observed after the second dose (OR = 0.67; 95% CI 0.51–0.89), and for CoronaVac3CL and Gam-COVID-Vac, protective outcomes were observed after the second dose. The Ad5-nCoV vaccine did not prevent symptomatic COVID-19 (Table 5). Multivariate analysis was performed by type of vaccine, and in all cases, being a health worker remained a protective

**Table 5. Vaccine protection to avoid getting sick from COVID-19.**

| Vaccine | All vaccinated | | 1 dose | | 2 dose | | Ideal interdose period | |
|---|---|---|---|---|---|---|---|---|
| | OR | CI95% | OR | CI95% | OR | CI95% | OR | CI95% |
| **Unvaccinated** | 1 | - | 1 | - | 1 | - | 1 | - |
| **All vaccines** | 0.61 | 0.57–0.66 | 0.9 | 0.83–0.97 | 0.31 | 0.28–0.35 | 0.7 | 0.56–0.86 |
| BNT162b2 mRNA | 0.35 | 0.32–0.39 | 0.61 | 0.53–0.70 | 0.18 | 0.15–0.21 | 0.21 | 0.12–0.39 |
| ChAdOx1 | 1.11 | 0.97–1.27 | 1.31 | 1.12–1.52 | 0.67 | 0.51–0.89 | 0.61 | 0.36–1.01 |
| CoronaVac3CL | 0.93 | 0.77–1.12 | 1.06 | 0.83–1.36 | 0.77 | 0.57–1.03 | 0.44 | 0.20–0.88 |
| Ad5-nCoV | 1.26 | 0.93–1.69 | 1.26 | 0.93–1.70 | - | - | 1.26 | 0.93–1.70 |
| Gam-COVID-Vac | 1.19 | 0.87–1.63 | 1.53 | 1.04–2.25 | 0.69 | 0.38–1.23 | 0.75 | 0.19–2.60 |
| **EFFECTIVENESS OF THE VACCINE TO AVOID GETTING SICK FROM COVID-19** | | | | | | | | |
| Vaccine | All vaccinated | | 1 dose | | 2 dose | | Ideal interdose period | |
| | VE | CI95% | VE | CI95% | VE | CI95% | VE | CI95% |
| **All vaccines** | 39 | 34;43 | 10 | 3;17 | 69 | 65;72 | 30 | 14;44 |
| BNT162b2 mRNA | 65 | 61;68 | 39 | 30;47 | 82 | 79;85 | 79 | 61;88 |
| ChAdOx1 | -11 | -27;3 | -31 | -52;-12 | 33 | 11;49 | 39 | -1;64 |
| CoronaVac3CL | 7 | -12;23 | -6 | -36;17 | 23 | -3;43 | 56 | 12;80 |
| Ad5-nCoV | -26 | -69;7 | -26 | -70;7 | - | -;- | -26 | -70;7 |
| Gam-COVID-Vac | -19 | -63;13 | -53 | -125;-4 | 31 | -23;62 | 25 | -160;81 |

factor (OR = 0.40, 95% CI 0.38–0.42). For comorbidities, two analyses were performed: chronic comorbidity (systemic arterial hypertension, diabetes mellitus, obesity, COPD, asthma and pulmonary tuberculosis) and severe comorbidity (cardiovascular pathology, cancer, renal pathology, liver pathology, HIV infection, hemolytic anemia and neurological pathology); severe comorbidities were a protective factor (OR = 0.44, 95% CI 0.41–0.45). The BNT162b2 mRNA vaccine remained an important protective factor against COVID-19 (OR = 0.30, 95% CI 0.22–0.32), followed by ChAdOx1 (OR = 0.70, 95% CI 0.55–0.97). Ad5-nCoV, again, was not associated with protection (S1 File). The highest VE was observed for BNT162b2 mRNA (82%), followed by ChAdOx1 (33%), when the second dose was included in the crude analysis. Based on logistic regression, BNT162b2 mRNA had a VE of 70%, and ChAdOx1 had a VE of 30%.

A total of 23,224 cases were hospitalized, of whom 12,327 died (lethality, 53%). Deaths occurred mainly among men (59.54%, 7,340). The average age of the deceased patients was 65 years (range = 18), while that of the survivors was 57 years (range = 18). A total of 3.29% (406) of the deceased cases were vaccinated against COVID-19: BNT162b2 mRNA (131), ChAdOx1 (120), CoronaVac3CL (56), Gam-COVID-Vac (26), Ad5-nCoV (18), Ad26 (2) and BBIBP-CorV (1). A total of 23.4% (95) of the deceased cases had received 2 doses. The percentage of survivors was 18.3% (84) ($X^2$ = 3.83, p = 0.05). (S1 File). Age greater than 64 years was found to be a risk factor for death (OR = 8.27, 95% CI 4.51–15.15). Being a health worker was a protective factor (OR = 0.36, 95% CI 0.28–0.47). Among comorbidities, hemolytic anemia was the main risk factor for death (OR = 2.9; 95% CI 1.18–7.81). Vaccination provided a protective effect against mortality (OR = 76, 95% CI 0.66–0.87). In relation to the type of vaccine, Ad5-nCoV was associated with the greatest reduction in risk (OR = 0.39, 95% CI 0.22–0.68), followed by ChAdOx1 (OR = 0.66, 95% CI 0.52–0.84). In the multivariate analysis, both BNT162b2 mRNA (OR = 0.71, 95% CI 0.54–0.92) and CoronaVac3CL (OR = 0.61, 95% CI 0.41–0.92) had protective effects. In addition, ChAdOx1 and Ad5-nCoV were again associated with low risk (OR = 0.42, 95% CI 0.23–0.77 and OR = 0.62, 95% CI 0.47–0.80, respectively). In the same model, a history of endotracheal intubation remained a risk factor for death (OR = 13.2, 95% CI 10.9–16.0), being a health worker remained a protective factor (OR = 0.49, 95% CI 0.38–0.64). VE, i.e. the prevention of death in patients with COVID-19, was 35% overall. VE was highest for the Ad5-nCoV vaccine (58%), followed by CoronaVac3CL (39%). In the survival analysis, a difference in protective effects was observed between vaccinated and unvaccinated people, mainly 1 month after vaccination. The effects of the ChAdOx1 and Ad5-nCoV vaccines were the most evident (Figs 1 and 2).

## Discussion

The epidemiological surveillance system database was analyzed to determine the effectiveness of biologicals through a population-based case-control study of people with social security. The results revealed that all vaccines confer, to a lesser or greater extent, protection against infection and death. Likewise, completing both doses confers greater protection. On December 31, 2020, the WHO authorized the emergency use of the BNT162b2 mRNA vaccine for SARS-CoV2; the United Kingdom had already authorized its use, and on December 8, the first vaccine was administered [21]. The IMSS, as the main institution of health and social security services in Mexico, has an information system for the detection of respiratory infections that was established because of the 2009 influenza A(H1N1)pdm09 pandemic [22] and has been used for the epidemiological surveillance of SARS-CoV2. Similarly, the IMSS has adhered to the national sectorial vaccination strategy to protect the population against COVID-19. Two biological were predominant between the cases and controls in the present study, with both

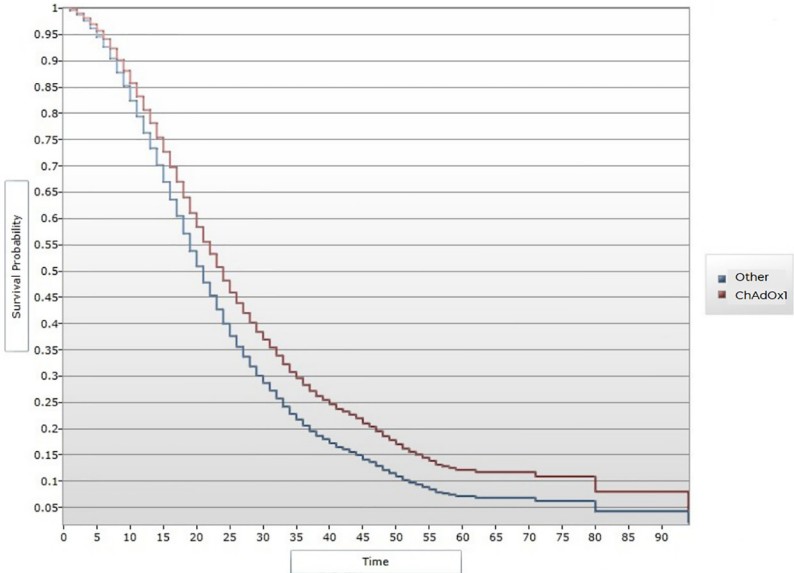

**Fig 1. Kaplan-Meier curve and vaccine efficacy over time after ChAdOx1.**

requiring two doses but from different manufacturers (mRNA and another vaccine vector). In a VE study that included 11,636 participants, the ChAdOx1 nCoV-19 vaccine prevented infection in 70.4% of participants 14 days after the second dose. Similarly, efficacy 21 days after the first dose conferred some protection against infection (58.9% [1.0–82.9]) [23]. Another study determined that the estimated efficacy, i.e., prevention of symptomatic infection, was 62.1% (95% CI 41.0–75.7) [24]. For biological mRNA, a meta-analysis of the BNT162b2 mRNA

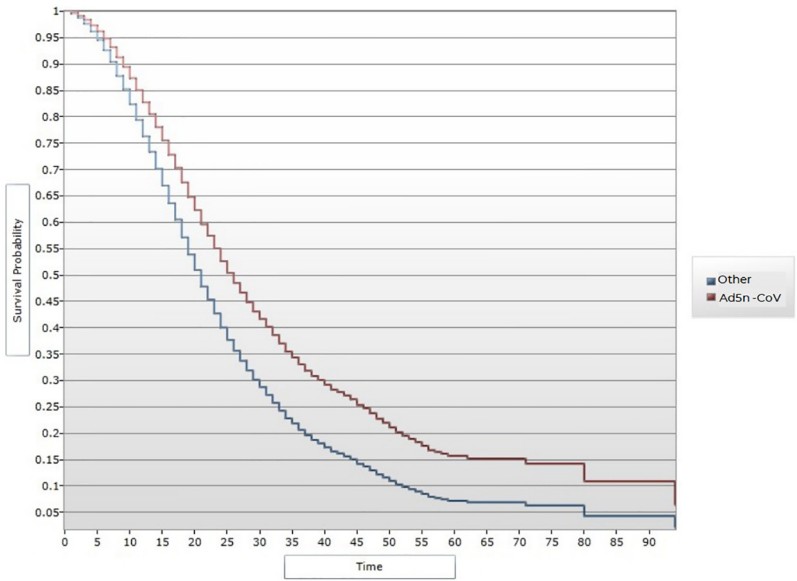

**Fig 2. Kaplan-Meier curve and vaccine efficacy over time after Ad5-nCoV.**

vaccine revealed that the protection against symptomatic infection with the first dose at 14 days was 53% (95% CI 32–68%), increasing to 95% after the second dose [25]. Previously, when the immune response conferred by a single dose is nearly or as robust as that following two doses, the incidence rate can be substantially reduced by delaying the second dose. Furthermore, it has been documented that the BNT162b2 vaccine elicits strong antibody responses. One week after the second dose, the neutralization titers of 50% of the geometric mean of SARS-CoV-2 in serum were up to 3.3 times higher than those observed in samples from individuals who had recovered from COVID-19 [26]. However, age influences immunity; the lower frequencies of neutralizing antibodies after vaccination with BNT162b2 in the older adult population may indicate a requirement for earlier revaccination [27]. In the present study, people aged 45 to 65 years had a higher risk of symptomatic COVID-19 and death among older adults, regardless of the type of vaccine administered.

The efficacy of vaccines diminishes with the appearance of viral variants, mainly with those of concern [28–31]. Viral variants had substantial impacts at the end of 2020, and henceforth, they have become worrisome worldwide. Although it was not possible to analyze variants in detail, when considering the context of the epidemiological behavior of COVID-19 in Mexico, there were two waves of infection in 2021: one at the beginning of the year that was surpassed by another in the middle of the year. Similarly, the behavior of the variants in Mexico changed in 2021; for example, variant B.1.1.519, present since the beginning of the year, predominated during May and peaked in June; however, the delta variant predominated between July and August [31], related to the increase in cases in the country, generating a third epidemic wave [2]. For those months, the most infected population group was those who were 40 to 49 years of age, a population that is economically active and highly mobile in the public environment. Across time, protective antibody levels progressively decrease [32, 33], and that protection is greater with complete schedules [33]. Vaccination rates have been slow [34] because of insufficient strategies and focused selection [35, 36].

In Mexico, a study of antibody generation after vaccination with Ad5-nCoV and BNT162b2 has been published [37]; among the participants, there were three primary comorbidities (systemic arterial hypertension, diabetes and obesity), differing only in their percentages, and 20% of participants had a history of allergies, a pathology not considered in this present study, given that allergies are not tracked in the epidemiological surveillance system. Regarding immunology, after the administration of the second dose, there was an elevation of IgG antibodies, an increase that was more evident for the BNT162b2 vaccine; antibodies were detected in 80.3% of the participants, increasing to 98.36% after the second dose, an effect that was more evident in women. In fact, in the present study, there were fewer women than men among the cases, and women accounted for fewer deaths, findings consistent with those of a previous study [38]. Likewise, the results demonstrate the general VE to protect against COVID-19 (69%), especially after a second dose; however, the BNT162b2 vaccine performed best, with a VE of 82% (39% VE after the first dose with); however, these results contrast with those at the population level in Spain, where there was greater prevention of symptomatic infection (69% after the first dose, 78.7% after the second dose, and 87.8% overall) [19]. One point to consider is that the cases and controls were taken from the same population with initial respiratory symptoms; however, the controls were negative in the RT-qPCR tests, so the answers to the interrogation had the same opportunity to be identified.

Based on the crude analysis and adjusted analysis, the risk of dying from COVID-19 was 35% and 12%, respectively. These values are lower than those reported in the Spanish study [19]. The Ad5-nCoV vaccine had the best VE (58% in the crude analysis and 30% in the adjusted analysis), followed by the ChAdOx1 vaccine (38%). The differences between the two studies may be due to the government vaccination strategy. For example, in Spain, after the

authorization of the first vaccine in the European Union, rollout began at the end of December (as in Mexico). However, the target groups for the first stage of vaccination were people at greater risk of exposure and those who more vulnerable, in addition to all health personnel (without discriminating for any condition) and noninstitutionalized people who were highly dependent on others for care. The difference may also be due to the type of vaccine available; the Pfizer/BioNTech is available in Mexico, but the Moderna vaccine is not [19]; the latter has an efficacy of 94.1% for infection 14 days after the first dose, with the same efficacy for the B.1.1.7 and 501Y.V2 variants [39]. So, when new cases activity and positivity are high, vaccinal immunity will be central to efforts to attain community immunity and prevent local spread and outbreaks; one or two doses immunity has waned will likely be able to be vaccinated again before infection, and so we incorporated re-vaccination of these individuals.

Working in a health area proved to be a protective factor, both for symptomatic disease and death. In this sense, the learning curve for health personnel has been steep since the beginning of the pandemic, mainly regarding the use of personal protective equipment, and despite having longer exposure times to patients with confirmed cases [40], health professionals are more aware of their care and among the first to be vaccinated. A single-center case-control study showed that the VE in health personnel who received BNT162b2 was 74.6% (95% CI: 38.4–89.5) [41], which is higher than the estimated value after a first dose for that biological, probably because of the estimation period for VE and the study population. Additionally, suffering from a severe concomitant disease, such as hemolytic anemia, cardiovascular disease or cancer, had a protective effect, which can be explained as follows: 1) the diseases may be underreported in the epidemiological surveillance system (primarily cardiovascular disease), 2) people with greater risk of complications could have a greater awareness in the application of preventive measures for their personal care, 3) specific medical units were assigned for the care of people with COVID-19 and 4) rapid tests were inserted into the protocol for people who underwent surgical events or invasive airway procedures [42].

Among other factors studied, endotracheal intubation was a risk factor for dying from COVID-19, explained by the clinical severity of the patient. Admission to intensive care was a protective factor, which can be explained by the better management of mechanical ventilation in these departments; however, intubation outside these departments may be executed improperly, an occurrence that could have increased due to the need of adding additional beds in other departments for the care of people infected by SARS-CoV2 [43].

Measuring the effectiveness of a vaccine is somewhat complex because the conditions in which they are tested in clinical trials can vary compared to those for an open population. The goal is to reduce disease through vaccination in a population under uncontrolled conditions; however, the efficacy values for clinical trials can vary: the duration of immunity relative to efficacy, the speed of acquisition/distribution/administration of the biological agent, the endpoint to be evaluated, circulating viral variants, the number of administered doses, prioritization of the target population, the use of heterologous regimens; and the degree of compliance with individual and collective prevention measures. The present study had the following strengths: 1) population based with the inclusion of various sites in the country, 2) standardized registration and coverage in all medical units, 3) RT–qPCR-verified infection, 4) VE evaluation of various biologicals during a pandemic wave, 5) number of records included and 6) inclusion of associated risk factors; therefore, the main attribute was that this was an open population study. The most important limitation was the source of the vaccine was self-reported; however, the data was obtained from an already structured information system, so it was not possible to analyze other variables of interest. and also there was differential bias between cases and controls; for example, 1) high-risk groups (people most

likely to present serious complications, such as older adults) or people with higher systematic exposure to the virus (such as health personnel) are more likely to be vaccinated and are prioritized in the vaccination scheme implemented in Mexico, 2) people included in July had received only a single dose of vaccine, 3) the effect of VE did not consider the history of a previous COVID-19 infection among the cases or the controls, 4) only homologous vaccination against COVID-19 was considered, 5) given the vaccination schedule by stages based on age group and place of residence, there may be different probabilities of being vaccinated and 6) VE for ICU admission or re infections were not included. Another point to consider is that the published studies on effectiveness are mainly clinical trials. It is still early to determine the effectiveness of biological agents only slightly more than half a year after vaccination; however, it has been announced that a third dose is required for certain populations [44] and that a second dose is recommended for those who only had a single dose [45]. However, the present study provides VE by the type of biological agent using the endpoints of symptomatic infection and death based on data in an epidemiological surveillance system 6 months after starting vaccination in Mexico. Likewise, it is one of the first studies in Mexico that contributes to a better understanding of this phenomenon. BNT162-2 and ChAdOx1 have been found to be efficacious against symptomatic COVID-19 and Ad5-nCoV for death in this interim analysis, so that this positive results presented here support regulatory submissions for conditional or emergency use vaccines in Mexico. The differences between the VE found also depend on vaccination schemes in each country; one of them, to extend the period between the first and second dose. Although it is already known that vaccination against COVID-19 reveals that our findings of this study also make clear that many more lives could have been saved, and will be saved; [46] not only for the population to know that information; but also for the health authorities when vaccination had only a mild effect on the substantial reduction in transmission rate and mortality/fatality rates when vaccination coverage are low or delayed [47]. Estimates of VE in the prevention of COVID-19 is essential because they reflect real-world challenges, such as vaccine program logistics and later epidemiological behavior change. Finally, in the context of pandemic and the continuous emergence of SARS-CoV-2 variants and sublineages, accelerating vaccination and improving vaccination coverage is still the most important and urgent matter, but also, to continue with individual and collective preventive measures, to reduce transmission because the appearance of new variants and lineages of the virus continues to be latent, which seems to be an endless race for the development of new vaccines in the face of said genomic changes of the virus.

## Conclusions

In summary, the impact of vaccines on the prevention of COVID-19 and the need to apply a second dose for greater efficacy, even a booster dose or second dose, depending on the type of biological agent, has already been recommended. The BNT162b2 mRNA vaccine presented the best VE regarding disease prevention, but Ad5-nCoV had the best efficacy regarding the prevention of death. Despite being vaccinated, it is necessary to continue with protective measures (such as masking) as well as to continue incentivizing the population to comply with the vaccination campaign against COVID-19.

## Supporting information

**S1 File.**
(XLSX)

## Acknowledgments

The authors thank all the participants in the viral respiratory disease epidemiological surveillance system, including treating physicians, epidemiologists, laboratory chemists, and nursing staff.

## Author Contributions

**Conceptualization:** Porfirio Felipe Hernández Bautista, Concepción Grajales Muñiz, Teresita Rojas Mendoza, Clara Esperanza Santacruz Tinoco.

**Formal analysis:** Porfirio Felipe Hernández Bautista.

**Investigation:** Alfonso Vallejos Parás, Clara Esperanza Santacruz Tinoco.

**Methodology:** Porfirio Felipe Hernández Bautista, Alfonso Vallejos Parás.

**Project administration:** Nancy Sandoval Gutiérrez.

**Resources:** Julio Elias Alvarado Yaah, Yu Mei Anguiano Hernández.

**Supervision:** David Alejandro Cabrera Gaytán.

**Visualization:** Teresita Rojas Mendoza.

**Writing – original draft:** Porfirio Felipe Hernández Bautista, David Alejandro Cabrera Gaytán.

**Writing – review & editing:** David Alejandro Cabrera Gaytán, Leticia Jaimes Betancourt.

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
