## [Decision Letter · Decision Letter 0]

20 Jun 2022

PONE-D-22-05555Impact of vaccination on infection or death from COVID-19 in individuals with laboratory-confirmed cases: case-control study.PLOS ONE

Dear Dr. David Cabrera Gaytan, 

Thank you for submitting your manuscript to PLOS ONE. After careful consideration, we feel that it has merit but does not fully meet PLOS ONE’s publication criteria as it currently stands. Therefore, we invite you to submit a revised version of the manuscript that addresses the points raised during the review process.

ACADEMIC EDITOR: Follow the reviewers corrections and recommendations ==============================

We look forward to receiving your revised manuscript.

Kind regards,

Shawky M Aboelhadid, PhD

Academic Editor

PLOS ONE

**Journal requirements:**

2. "In your ethics statement in the manuscript and in the online submission form, please provide additional information about the patient records used in your retrospective study. Specifically, please ensure that you have discussed whether all data were fully anonymized before you accessed them.

5. Please include a copy of Tables 1 to 5 which you refer to in your manuscript.

Reviewers' comments:

Reviewer's Responses to Questions

**Comments to the Author**

1. Is the manuscript technically sound, and do the data support the conclusions?

Reviewer #1: Partly

Reviewer #2: Partly

2. Has the statistical analysis been performed appropriately and rigorously? 

Reviewer #1: Yes

Reviewer #2: Yes

3. Have the authors made all data underlying the findings in their manuscript fully available?

Reviewer #1: No

Reviewer #2: Yes

4. Is the manuscript presented in an intelligible fashion and written in standard English?

Reviewer #1: Yes

Reviewer #2: Yes

5. Review Comments to the Author

Reviewer #1: The goal of this study is relevant. Although that are a few points that require attention and better discussion.

1. Better graphic representation of the results is needed.

2. The time after vaccination should be taken into consideration since studies have shown that this could be a limiting factor for protection. Thus, comparing people vaccinated in different moments could be a problem.

3. The fact that the vaccine source was self-reported makes the conclusions weaker. Is is possible to confirm the source of vaccination?

4. The differential bias between cases and controls should be taken into consideration and better discussed in the manuscript.

5. The figure need to have better quality and higher resolution.

Reviewer #2: Article with a very current theme, but needs adjustments for publication in PLOS ONE.

ABSTRACT / Methods – “... those with a positive result were considered cases, and

60 those with a negative result were considered controls.” - What is the ratio between cases and controls? This information needs to be detailed in the summary.

ABSTRACT / Results – “of whom 40,192 were considered 65 cases and 54,224 controls;” - Would the 1:1 ratio be able to identify the differences? Explain.

“3,781 (4.00%) had been vaccinated against COVID-19.” - What is the proportion of vaccination in cases and controls? Explain.

“(OR = 76; 95% CI 69 0.66-0.87).” - OR = 76 – Review this information.

PURPOSE – Adjust according to the Introduction. Review the Introduction.

METHODS

Operational definitions -> “...at least 15 days before the onset of symptoms” - From the second dose? From the first dose? It would impact.

RESULTS

It is important for the table to show that the groups are homogeneous, but characteristic. Otherwise (heterogeneous), there could be a bias.

DISCUSSION

It is well structured, however very directed to the basic field of science.

Restructuring to respond to the objective from a public health perspective: What was the impact of vaccination on population health? I suggest better highlighting the benefits of vaccines for the health of the population.

What are the limitations of the study? Explain better.

6. PLOS authors have the option to publish the peer review history of their article (what does this mean?). If published, this will include your full peer review and any attached files.

Reviewer #1: **Yes: **Juliana P B Menezes

Reviewer #2: **Yes: **FERNANDA PENIDO MATOZINHOS

---

## [Author Response · Author response to Decision Letter 0]

30 Dec 2022

General.

A = We did the changes.

2. "In your ethics statement in the manuscript and in the online submission form, please provide additional information about the patient records used in your retrospective study. Specifically, please ensure that you have discussed whether all data were fully anonymized before you accessed them.

A = We included the Ethical approval. Also, de basis was fully anonymized.

A = Available on figshare: 10.6084/m9.figshare.21691229

4. Please include captions for your Supporting Information files at the end of your manuscript, and update any in-text citations to match accordingly.

A = We did the changes.

5. Please include a copy of Tables 1 to 5 which you refer to in your manuscript.

 A = the tables were loaded, as well as the supporting information tables.

Review Comments to the Author.

Reviewer #1: The goal of this study is relevant. Although that are a few points that require attention and better discussion.

1. Better graphic representation of the results is needed.

A = Graphics files have been modified.

2. The time after vaccination should be taken into consideration since studies have shown that this could be a limiting factor for protection. Thus, comparing people vaccinated in different moments could be a problem.

A = It was omitted to include in the operational definition that the time is from the second dose or from the only dose, depending on the biologic. The timing of vaccination was not possible to control. The strategy in Mexico was to start by older adults to younger.

3. The fact that the vaccine source was self-reported makes the conclusions weaker. Is possible to confirm the source of vaccination?

A = The information was obtained from an already structured information system, so it was not possible to analyze other variables of interest. In Reviewer 2's comment on limitations, the statement was modified to address both comments.

4. The differential bias between cases and controls should be taken into consideration and better discussed in the manuscript.

A = The cases and controls were taken from the same population with initial respiratory symptoms; however, the controls were negative in the PCR tests, so the answers to the questioning had the same opportunity to be identified.

5. The figure need to have better quality and higher resolution.

A = Graphics files have been modified.

Reviewer #2: Article with a very current theme, but needs adjustments for publication in PLOS ONE.

1. ABSTRACT / Methods – “... those with a positive result were considered cases, and 60 those with a negative result were considered controls.” - What is the ratio between cases and controls? This information needs to be detailed in the summary.

A = The ratio between cases and controls was 1:1.3.

2. ABSTRACT / Results – “of whom 40,192 were considered 65 cases and 54,224 controls;” - Would the 1:1 ratio be able to identify the differences? Explain.

A = The ratio between cases and controls was 1:1.3.

3. “3,781 (4.00%) had been vaccinated against COVID-19.” - What is the proportion of vaccination in cases and controls? Explain.

A = The vaccination proportion in cases was 3.35% and controls 5.31%. The low vaccination coverage is due to the fact that the study was carried out when the application of vaccines was just beginning. However, a lower proportion of vaccinated cases is observed.

4. “(OR = 76; 95% CI 69 0.66-0.87).” - OR = 76 – Review this information.

A = With the application of the vaccine, there was a protective effect against mortality (OR = 0.76; 95% CI 0.66-0.87).

5. METHODS. Operational definitions -> “...at least 15 days before the onset of symptoms” - From the second dose? From the first dose? It would impact.

A = From the second dose.

6. RESULTS. It is important for the table to show that the groups are homogeneous, but characteristic. Otherwise (heterogeneous), there could be a bias.

A = The study included the universe of work technically to avoid any type of selection bias and logistic regression was performed to control possible confounders.

7. DISCUSSION. It is well structured, however very directed to the basic field of science. Restructuring to respond to the objective from a public health perspective: What was the impact of vaccination on population health? I suggest better highlighting the benefits of vaccines for the health of the population.

A = It is now known from published studies that counties/countries with higher vaccination coverage had fewer COVID-19 cases and deaths per head of population and the measured effectiveness in counties/countries with high vaccine coverage was reassuringly large and the vaccination benefits whole communities, and indeed it does when coverage is high. Vaccine rollout began in Mexico with older and otherwise vulnerable people and progressively included younger and less vulnerable people; although we did not investigate the effect of vaccination by age, but our findings of this study also make clear that many more lives could have been saved, and will be saved. That’s why we added: “Although it is already known that vaccination against COVID-19 reveals that our findings of this study also make clear that many more lives could have been saved, and will be saved;42 not only for the population to know that information; but also for the health authorities when vaccination had only a mild effect on the substantial reduction in transmission rate and mortality/fatality rates when vaccination coverage are low or delayed.43”

8. What are the limitations of the study? Explain better.

A = The limitations of the study are described in the discussion; however, detailing the statement with that subtitle was omitted. has been added to the manuscript. “The most important limitation was the source of the vaccine was self-reported; however, the data was obtained from an already structured information system, so it was not possible to analyze other variables of interest., and there was differential bias between cases and controls; for example, 1) high-risk groups (people…

---

## [Decision Letter · Decision Letter 1]

1 Feb 2023

PONE-D-22-05555R1Impact of vaccination on infection or death from COVID-19 in individuals with laboratory-confirmed cases: case-control study.PLOS ONE

Dear Dr. Gaytan, 

Thank you for submitting your manuscript to PLOS ONE. After careful consideration, we feel that it has merit but does not fully meet PLOS ONE’s publication criteria as it currently stands. Therefore, we invite you to submit a revised version of the manuscript that addresses the points raised during the review process.

ACADEMIC EDITOR: The reviewers have some comments and suggestions for improving the manuscript, Authors need to revise the manuscript according these comments.

We look forward to receiving your revised manuscript.

Kind regards,

Shawky M Aboelhadid, PhD

Academic Editor

PLOS ONE

Journal Requirements:

Reviewers' comments:

Reviewer's Responses to Questions

**Comments to the Author**

1. If the authors have adequately addressed your comments raised in a previous round of review and you feel that this manuscript is now acceptable for publication, you may indicate that here to bypass the “Comments to the Author” section, enter your conflict of interest statement in the “Confidential to Editor” section, and submit your "Accept" recommendation.

Reviewer #1: All comments have been addressed

Reviewer #2: All comments have been addressed

2. Is the manuscript technically sound, and do the data support the conclusions?

Reviewer #1: Partly

Reviewer #2: Yes

3. Has the statistical analysis been performed appropriately and rigorously? 

Reviewer #1: Yes

Reviewer #2: Yes

4. Have the authors made all data underlying the findings in their manuscript fully available?

Reviewer #1: Yes

Reviewer #2: Yes

5. Is the manuscript presented in an intelligible fashion and written in standard English?

Reviewer #1: Yes

Reviewer #2: Yes

6. Review Comments to the Author

Reviewer #1: Although the manuscript reports important data about a relevant topic, some points should still be considered.

1. Regarding the time after vaccination, although it was impossible to control, it should be considered in the analysis.

2. The fact that the vaccine source was self-reported compromises the conclusions. Is that possible to access a governmental system to check that? Is that available in Mexico?

3. In Methods: when the authors mention that "at least 15 days before the onset of symptoms", the information that it is from the second dose should be added.

4. The discussion should still be better restructured, considering the importance of vaccination to the population.

Reviewer #2: Dear Editor and Authors,

Thank you for the opportunity to review this manuscript. I am grateful for the invitation.

After careful consideration, I feel that the manuscript has a satisfying result and explores a very important topic, but there few weaknesses in the Introduction, methods and discussion:

The introduction is very concise. I suggest better exploring epidemiological data, including recent international studies, as well as inserting a more robust justification for the study.

The methods are well described, with rigor in data analysis.

Explain the period of data collection in the system.

In the discussion and conclusions: include how the results of the study can be extrapolated, in order to explain the relevance of this paper.

Kind regards,

7. PLOS authors have the option to publish the peer review history of their article (what does this mean?). If published, this will include your full peer review and any attached files.

Reviewer #1: No

Reviewer #2: **Yes: **Fernanda Penido Matozinhos

While revising your submission, please upload your figure files to the Preflight Analysis and Conversion Engine (PACE) digital diagnostic tool, https://pacev2.apexcovantage.com/. PACE helps ensure that figures meet PLOS requirements. To use PACE, you must first register as a user. Registration is free. Then, login and navigate to the UPLOAD tab, where you will find detailed instructions on how to use the tool. If you encounter any issues or have any questions when using PACE, please email PLOS at figures@plos.org. Please note that Supporting Information files do not need this step.<quillbot-extension-portal></quillbot-extension-portal>

---

## [Author Response · Author response to Decision Letter 1]

23 Feb 2023

Reviewer #1: Although the manuscript reports important data about a relevant topic, some points should still be considered.

1. Regarding the time after vaccination, although it was impossible to control, it should be considered in the analysis.

A = Table 1. “Deaths and survivors, main characteristics”. We added “Onset after vaccination*”

COVID-19 cases had a median number of fewer days after being vaccinated compared to controls 20 vs 42, p<0.000, while in deceased and survivors of COVID-19 there was no statistically significant difference 19 vs 20, p= 0.4181. 

2. The fact that the vaccine source was self-reported compromises the conclusions. Is that possible to access a governmental system to check that? Is that available in Mexico?

A = Public information is numerical reports. Although the available database is counted, but it is impossible to identify the people notified in the epidemiological surveillance system.

3. In Methods: when the authors mention that "at least 15 days before the onset of symptoms", the information that it is from the second dose should be added.

A = This observation was added to the methodology.

4. The discussion should still be better restructured, considering the importance of vaccination to the population.

A = It has been modified and paragraphs have been added to the discussion.

Reviewer #2: Dear Editor and Authors,

Thank you for the opportunity to review this manuscript. I am grateful for the invitation.

After careful consideration, I feel that the manuscript has a satisfying result and explores a very important topic, but there few weaknesses in the Introduction, methods and discussion:

The introduction is very concise. I suggest better exploring epidemiological data, including recent international studies, as well as inserting a more robust justification for the study.

A = The introduction was modified, in order to strengthen the justification of the study.

The methods are well described, with rigor in data analysis.

A = Thanks for the observation.

Explain the period of data collection in the system.

A = The source of information was the epidemiological surveillance system of COVID-19; therefore, the time of registration of each person was when the medical unit notified the case. For the data collection, for the purposes of the study it was one month later (July 22, 2021).

In the discussion and conclusions: include how the results of the study can be extrapolated, in order to explain the relevance of this paper.

A = It has been modified and paragraphs have been added to the discussion.

---

## [Decision Letter · Decision Letter 2]

29 Mar 2023

Impact of vaccination on infection or death from COVID-19 in individuals with laboratory-confirmed cases: case-control study.

PONE-D-22-05555R2

Dear Dr. Alejandro, 

We’re pleased to inform you that your manuscript has been judged scientifically suitable for publication and will be formally accepted for publication once it meets all outstanding technical requirements.

Kind regards,

Shawky M Aboelhadid, PhD

Academic Editor

PLOS ONE

Additional Editor Comments (optional):

Reviewers' comments:

Reviewer's Responses to Questions

**Comments to the Author**

1. If the authors have adequately addressed your comments raised in a previous round of review and you feel that this manuscript is now acceptable for publication, you may indicate that here to bypass the “Comments to the Author” section, enter your conflict of interest statement in the “Confidential to Editor” section, and submit your "Accept" recommendation.

Reviewer #1: All comments have been addressed

Reviewer #2: All comments have been addressed

2. Is the manuscript technically sound, and do the data support the conclusions?

Reviewer #1: Yes

Reviewer #2: Yes

3. Has the statistical analysis been performed appropriately and rigorously? 

Reviewer #1: Yes

Reviewer #2: Yes

4. Have the authors made all data underlying the findings in their manuscript fully available?

Reviewer #1: Yes

Reviewer #2: Yes

5. Is the manuscript presented in an intelligible fashion and written in standard English?

Reviewer #1: Yes

Reviewer #2: Yes

6. Review Comments to the Author

Reviewer #1: All the comments made were addressed by the authors. The manuscript was improved and in my opinion, can be accepted for publication.

Reviewer #2: Dear Authors and Editor,

Thank you for the opportunity to review this manuscript. I am grateful for the invitation.

After careful consideration, I feel the manuscript explores a very important topic. The questions were responded and modifications in the text made the manuscript come to a satisfying result.

Kind regards.

7. PLOS authors have the option to publish the peer review history of their article (what does this mean?). If published, this will include your full peer review and any attached files.

Reviewer #1: No

Reviewer #2: **Yes: **Fernanda Penido Matozinhos

<quillbot-extension-portal></quillbot-extension-portal>

---

## [Editor Report · Acceptance letter]

25 Apr 2023

PONE-D-22-05555R2 

Impact of vaccination on infection or death from COVID-19 in individuals with laboratory-confirmed cases: case-control study. 

Dear Dr. Cabrera Gaytán:

I'm pleased to inform you that your manuscript has been deemed suitable for publication in PLOS ONE. Congratulations! Your manuscript is now with our production department. 

Kind regards, 

on behalf of

Professor Shawky M Aboelhadid 

Academic Editor

PLOS ONE